# Longitudinal Gait Analysis of a Transfemoral Amputee Patient: Single-Case Report from Socket-Type to Osseointegrated Prosthesis

**DOI:** 10.3390/s23084037

**Published:** 2023-04-17

**Authors:** Stefano Di Paolo, Giuseppe Barone, Domenico Alesi, Agostino Igor Mirulla, Emanuele Gruppioni, Stefano Zaffagnini, Laura Bragonzoni

**Affiliations:** 1Department for Life Quality Studies, University of Bologna, 47921 Rimini, Italy; 2II Orthopaedic and Traumatologic Clinic, IRCCS Istituto Ortopedico Rizzoli, 40136 Bologna, Italy; 3Department of Engineering, University of Palermo, 40126 Palermo, Italy; 4Istituto Nazionale Assicurazione Infortuni sul Lavoro (INAIL), Centro Protesi Inail, 40054 Vigorso di Budrio, Italy

**Keywords:** transfemoral amputation, osseointegration, socket-type, biomechanics, wearable sensors, gait analysis, gait symmetry, case report

## Abstract

The aim of the present case report was to provide a longitudinal functional assessment of a patient with transfemoral amputation from the preoperative status with socket-type prosthesis to one year after the osseointegration surgery. A 44 years-old male patient was scheduled for osseointegration surgery 17 years after transfemoral amputation. Gait analysis was performed through 15 wearable inertial sensors (MTw Awinda, Xsens) before surgery (patient wearing his standard socket-type prosthesis) and at 3-, 6-, and 12-month follow-ups after osseointegration. ANOVA in Statistical Parametric Mapping was used to assess the changes in amputee and sound limb hip and pelvis kinematics. The gait symmetry index progressively improved from the pre-op with socket-type (1.14) to the last follow-up (1.04). Step width after osseointegration surgery was half of the pre-op. Hip flexion-extension range significantly improved at follow-ups while frontal and transverse plane rotations decreased (*p* < 0.001). Pelvis anteversion, obliquity, and rotation also decreased over time (*p* < 0.001). Spatiotemporal and gait kinematics improved after osseointegration surgery. One year after surgery, symmetry indices were close to non-pathological gait and gait compensation was sensibly decreased. From a functional point of view, osseointegration surgery could be a valid solution in patients with transfemoral amputation facing issues with traditional socket-type prosthesis.

## 1. Introduction

A limb amputation is a life-changing event that limits a person’s independence, quality of life, and participation in everyday activities. In developed countries, the main cause of lower limb amputation is vascular disease, with diabetes mellitus accounting for two-thirds of all amputations; in developing countries, traumatic etiology related to occupational accidents, road trauma, and blast trauma in war situations is the most common cause [1]. Another important cause worldwide is the musculoskeletal tumor, especially in young males [2].

Although technological progress has allowed prosthesis customization according to a patient’s individual needs, difficulties in the fabrication and daily use of the socket remain very common among patients with lower limb amputation and represent the main cause of dissatisfaction. Pressure sores, skin abrasions due to friction, excessive skin sweating, changes in stump volume, lack of balance, and walking difficulties are very frequent problems among amputees, who abandon the use of the socket, reducing their overall quality of life [3,4,5,6,7]. In particular, amputees must modulate their kinematic pattern according to the musculature of the residual limb [2,6,8,9]. The gait is characterized by a poor hip flexion-extension of the amputated limb during the ipsilateral and contralateral heel strike and contralateral toe-off phases. In addition, the healthy limb has a greater anterior pelvic tilt and abduction, resulting in a limping gait [8]. The spatiotemporal gait parameters are usually asymmetrical and shifted towards the contralateral limb.

The osseointegration technique represents an alternative treatment for amputees with socket-related problems and low quality of life. This technique involves implanting a stem within the medullary canal of the amputated skeletal segment that extends outside the amputation stump. A prosthesis is then attached to the metal extension using a quick-release coupling system. These implants exploit the properties of the recipient’s bone to grow within the macro-porosity present on their external surface, a process that leads to osseointegration within the amputated bone [10]. The possibility of attaching the external prosthesis directly to the amputated skeletal segment has many advantages for the patient, including the elimination of the socket and the related skin problems, increased skeletal proprioception, and, theoretically, an improved gait cycle efficiency with better control of the stump [3,11,12,13,14,15,16].

Therefore, interest in this technique has grown considerably in recent years, and patients who are dissatisfied with their sockets are asking for such an innovative intervention to improve their quality of life and improve their functional levels [17,18].

However, only limited knowledge on gait pattern differences between socket and osseointegrated prostheses is available in the literature [3]. Previous studies were solely performed in a laboratory environment and focused on asymmetries in residual muscle forces and kinetics, despite indicating limited accuracy of the latter [8,19]. In particular, no studies reported longitudinal gait analysis of patients that moved from socket to osseointegrated prosthesis. Moreover, with the advent of wearable sensors and their ever wider applicability in clinical scenarios, it is desirable to provide kinematic data collected in more ecological environments with easy-to-use and clinically relevant technology [6].

The aim of the present case report was to provide a longitudinal functional assessment of a patient with transfemoral amputation from the preoperative status with socket prosthesis to one year after the osseointegration surgery by means of wearable inertial sensors. The hypothesis was that an osseointegrated prosthesis allows walking with greater limb symmetry and hip range of motion, and with a reduced limping pattern, compared to use of a socket-type prosthesis.

## 2. Materials and Methods

The present study investigated the gait kinematics of a participant with transfemoral amputation treated with osseointegration surgery at IRCCS Istituto Ortopedico Rizzoli of Bologna, Italy and rehabilitated at INAIL prosthetic center of Vigorso di Budrio, Italy. The study was approved by the Bioethical Committee of University of Bologna (ID protocol n° 25861). The patients signed informed consent before being enrolled in the study.

### 2.1. Case Presentation

The patient is a 44 years-old male who underwent a left transfemoral amputation following a motorbike accident in 2003 (Table 1). The patient had also suffered an amputation of the fifth finger of the left hand at the proximal interphalangeal level, a lacerated contusion wound in the volar region of the left hand, and a traumatic injury to the extensor muscles of the left hand. Ipsilateral hip movements were complete and pain-free. No other major comorbidities were present.

Despite the implementation of several prostheses, he was not fully satisfied with his socket, which caused decubitus at the ischial and gluteal level, pain at the level of the scar, excessive sweating, and easy fatigability when walking, with important limitations in the continuous and satisfactory use of the external prosthesis. He reported occasional paresthesia in the distal part of the stump, without a strength deficit.

After a preliminary evaluation by a multidisciplinary team composed of an orthopedist, physiatrist, prosthetic technician, and psychologist, an indication was given for revision surgery of the left transfemoral stump and implantation of an osseointegrated prosthesis, to allow the patient to use the external prosthesis without socket with probable resolution of skin problems, better muscle control of the stump and an overall improvement in quality of life.

In March 2021, the patient then underwent the first surgical step of stump revision and implantation of the osseointegrated intramedullary stem. After 6 weeks, the second surgical step of creating the skin stoma and implanting the transcutaneous double-cone adapter was performed. After a convalescence period of 15 days, the patient started the 17-day rehabilitation program.

Follow-up visits were performed one month after the 2nd surgical step and subsequently at 3, 6, and 12 months post-operatively (Figure 1). No adverse events were reported regarding the ostomy and the implant-related issues. The patient’s only complaint at the last follow-up relates to inflammation of the ostomy, which presents small bleeding after long walks (>2 km).

### 2.2. Surgical Technique

The implantation of the osseointegrated prosthesis for transfemoral amputation was performed in two distinct surgical steps, between which there were approximately 60 days: this interval allowed for the complete healing of the surgical scars and soft tissues and the initial osseointegration of the press-fit BADAL X Femur Implant OFI-C (OTN Implants, Netherlands).

In the first surgical step, a revision of the amputation stump was performed, regularizing the distal portion of the skeletal segment, and removing excess soft tissue. The intramedullary canal was prepared with flexible reamers and rasps of increasing size until the optimal press-fit was achieved. Prior to implantation of the definitive stem, four holes were drilled in the distal portion of the skeletal stump, through which the sutures for the distal myodesis were passed.

The definitive press-fit stem was then implanted, taking care not to create fractures, and the bone marrow tissue obtained from reaming the canal was applied to the distal interface of the prosthetic bone to promote osseointegration. A healing plug was inserted inside the housing of the future double-cone adapter to prevent the growth of fibrous tissue within it.

The site of the future ostomy was then identified with a k-wire and excess soft tissue was removed. The stump was sutured in layers and a compression bandage was applied to reduce postoperative swelling. After the first operation, the patient could not wear the prosthesis with socket and load so as not to interfere with the osseointegration process and because the shape of the stump was changed.

After at least 6 weeks, the second surgical step was performed. A special drill was used to create the skin ostomy. The healing plug was removed, the lodging on the distal portion of the stem was carefully washed out and the double-cone adapter of the planned length was implanted. The double-cone adapter was fixed with a morse taper system, secured with a screw.

Three weeks after the second operation, the patient began a 6-week rehabilitation program, with an intermediate break. The alignment of the external prosthesis was fine-tuned, and the patient started to regain the functions of balance, walking, and targeted muscle strengthening under the guidance of physiatrists and physiotherapists experienced in treating amputees (Figure 2).

### 2.3. Data Collection

Functional tests were performed the day before osseointegration surgery, i.e., the last day of socket-type prosthesis (ST Pre-op) and then at 3 months (OI FU 3M), 6 months (OI FU 6M), and 12 months (OI FU 1Y) after the second osseointegration surgical step. The first follow-up coincided with the rehabilitation clearance by the physiotherapists’ team.

The gait analysis was performed using a set of 15 wearable inertial sensors (MTw Awinda, Xsens Technologies, Enschede, The Netherlands) placed bilaterally on feet (back), shins (internally on the flat bone), thighs (laterally), arms (laterally), forearms (laterally), and shoulders (scapulae); one sensor was placed on the back of the pelvis (at L5 level), on the trunk (at xiphoid process), and the head. The sensors at the thigh and the shin were placed at the same height level for the two limbs, with the straps covering the stump at the thigh and the prosthetic knee at the shin for the amputee limb (Figure 3). The calibration of the sensors system was performed according to the manufacturer’s instructions and the accuracy of the calibration was verified accordingly.

A 20-m walk (roundtrip in a 10-m path) was carried out in the indoor hall of the hospital where the patient had undergone the two surgical steps and the acute postoperative phase, twice at a self-selected speed and twice at the maximum speed possible.

Questionnaires were submitted to the patient before the surgery, and then 3 and 12 months after surgery. The questionnaires included the Questionnaire for Persons with Transfemoral Amputation (Q-TFA) and the European Questionnaire 5-dimension, 5-level (EQ-5D-5L) [20,21,22]. Q-TFA is a validated self-reported measure developed to determine the health-related quality of life of a person with transfemoral amputation. EQ-5D-5L is a non-disease-specific instrument that provides a generic measure of health status by means of 5 dimensions with 5 levels of problems per dimension.

### 2.4. Data Analysis

Wearable sensors data were processed in the Gait Analysis Report in the Xsens Motion Cloud (https://www.xsens.com/motioncloud, accessed on 15 June 2022) where gait events were automatically identified. Joint angles were defined using the Euler sequence ZXY using sensors fusion algorithms as reported in the manufacturer’s description (https://www.xsens.com/hubfs/Downloads/usermanual/MVN_User_Manual.pdf, accessed on 15 June 2022) and further post-processed in a customized script in Matlab (The MathWorks, Natick, Massachusetts, MA, USA). All parameters were investigated according to the normalized gait cycle (0–100%). The following spatial parameters were assessed: speed, cadence, step length, and step width. The temporal parameters under investigation were the gait cycle percentage of stance and swing phase, single and double support phase, symmetry index (assessed as the ratio between sound and amputee limb stance phase [23]), and coefficient of variation (assessed as the percentage of the ratio between standard deviation and average of the stride time [24]). The symmetry index decreased as the symmetry increased (symmetry index = 1.00 means perfect symmetry [23], and a threshold of 2.6% for the coefficient of variation was identified in the literature to describe the pathological gait [24]. The 3D joint kinematics of the hip (amputee and sound limb), knee and ankle (sound limb), and pelvis were investigated.

Q-TFA is composed of 4 scores, ranking 0 to 100 points, reflecting prosthesis use, prosthetic mobility, prosthesis-related problems, and global health. A higher score corresponds to a better result, except for the score Problem where 100 is the worst result. EQ-5D-5L has 5 domains: mobility, self-care, usual activities, pain/discomfort, and anxiety/depression. Each of the domains measures health using five levels of severity: level 1 denotes no issue, while level 5 denotes the most severe restriction. The questionnaire also includes a “visual analog scale value set” question, scored 0 to 100, about the perceived general status of health.

### 2.5. Statistical Analysis

Continuous variables are presented as mean and standard deviation, while categorical variables are presented as sample size and percentages over the total. The Repeated measures ANOVA was used to assess the differences between the follow-ups for the continuous kinematic variables through the Random Field Theory according to Pataky et al. in Statistical Parametric Mapping 1D (spm1D) [25]. The Student’s t-test with Bonferroni correction for multiple comparisons was performed to compare each follow-up couple. Statistical tests were performed in Matlab. The post-hoc differences between the preoperative gait analysis (ST pre-op) and the first (OI FU 3M), intermediate (OI FU 6M), and last (OI FU 1Y) follow-ups are reported and further discussed. Differences between the groups are considered statistically significant if *p* < 0.05.

## 3. Results

Minimal differences in terms of speed and kinematics emerged between self-selected speed gait and fast gait. For conciseness, only fast gait results are presented in the main manuscript [26], while self-selected speed data are presented in Appendix A. The number of included amputee and sound limb gait cycles in the fast gait were 12, 14, 16, and 8 for ST pre-op, OI FU 3M, OI FU 6M, and OI FU 1Y, respectively.

### 3.1. Spatiotemporal Parameters

Gait speed was comparable among the follow-ups. Cadence decreased from ST pre-op to all the follow-ups. Step width decreased by one-half between ST pre-op and all the follow-ups for both the amputee and the sound limb (Table 2).

The side-to-side difference between the amputee and sound limb in stance and swing phase progressively decreased from the ST pre-op through all the follow-ups. The symmetry index was the highest in the ST pre-op (1.14, lowest symmetry) and the lowest in the last follow-up OI FU 1Y (1.04, highest symmetry). The coefficient of variation was the lowest in the last follow-up OI FU 1Y for both the amputee and the sound limb and was always below the pathological threshold [23] (Table 3). The side-to-side difference between the amputee and sound limb in the single support phase progressively decreased over time and the total percentage of the double support phase was up to double in the last follow-ups (highest in OI FU 6M) compared to the ST pre-op (Table 3).

### 3.2. Joint Kinematics

Hip flexion-extension differed significantly between the ST pre-op and the OI follow-up (*p* < 0.001): after 6 months from the surgery, a reduced flexion peak at foot contact and an increased extension peak at toe-off were noted for both the amputee and sound limbs (Figure 4 and Figure 5). In the amputee limb, hip abduction and hip internal rotation were also reduced from the preoperative to the 3M and 6M follow-ups, while peak abduction and rotation between pre-op and 1Y follow-up during the swing phase were comparable (Figure 4).

Pelvis kinematics significantly changed over time on the three planes: reduced anteversion during the entire gait cycle and reduced rotation and obliquity at amputee limb foot contact was noted (Figure 6 and Figure 7). Lower pelvis rotation at toe-off was also noted during the 1Y follow-up compared to the preoperative phase (*p* < 0.001).

Full descriptive peak gait analysis is presented in Table 4 and Table 5.

### 3.3. Questionnaires

Q-TFA shows improvements for all scores between pre-op, 3-month follow-up, and 1-year follow-up (Table 6). Only the Prosthetic Mobility score slightly decreased from 93.3 at 3-M to 92.2 at 1-Y.

EQ-5D-5L shows better results on Mobility, Self-care, Usual Activities, and Pain/Discomfort after the surgery. The domain Anxiety/Depression was at a minimum level of severity during pre-op and maintained the same value also during 3-M and 1-Y follow-up. No differences were found between all five domains between the 3-M and 1-Y follow-up. The General Health score increased strongly after the surgery and slightly increased between 3-M and 1-Y follow-up (Table 7).

## 4. Discussion

The most important finding of the present case report study was the strong difference in gait spatiotemporal parameters and joint kinematics in a transfemoral amputee patient after an osseointegration surgical treatment compared to a standard socket-type prosthesis. The patient exhibited a progressive improvement of the gait parameters over time and reached symmetry indices close to non-pathological gait in both fast and self-selected speed [23]. Typical gait patterns for transfemoral amputee patients, e.g., low hip flexion and limping, were significantly reduced from the first follow-up after surgery. Quality of life measures (Q-TFA and EQ-5D-5L [27]) also improved from pre-operation to follow-ups. Such findings suggest that the patient benefitted from the osseointegration surgery both in movement functionality and satisfaction (Figure 8). The present case report study represents the first longitudinal functional assessment of a transfemoral amputee patient treated with osseointegration during the first year after surgery including gait analysis conducted by means of wearable inertial sensors. Such an analysis might be of crucial importance in informing the design of future studies focusing on a practical and relevant clinical assessment of walking abilities of transfemoral amputee patients.

Spatiotemporal parameters sensibly improved after surgery. A stance phase close to 60% of the gait cycle was reached after surgery and is in line with previous studies comparing able bodied participants and patients with bone-anchored prosthesis [18]. Step width in both the amputee and sound limb was half of the pre-op already in the first follow-up and further decreased over time (Table 2). The wider base of support adopted by the patient with the socket-type prosthesis is commonly found in multiple gait diseases and is adopted to keep medio-lateral stability in the presence of poor motor control [28]. Excessive step width is also recognized as an important predictor of the risk of fall [29]. Despite a reduced step width, the patient did not demonstrate a reduced gait speed or step cadence, which remained comparable among the follow-ups (Table 2). A lower gait speed in the presence of a narrower base of support is commonly adopted by patients as a compensatory mechanism to avoid the risk of falling [30,31]. However, it should be noted that the patient’s gait speed and cadence were higher than the reference values for amputees reported in the literature and were close to normality already in the pre-op [23,32,33] The decreased base of support suggests that the patient gained greater confidence in the amputee limb compared to the socket-type prosthesis and could be related to improved sensorimotor control.

The gait symmetry index continuously improved during the longitudinal assessment (Table 3). The reduced differences between amputee and sound limb stance period indicated a more physiological gait achieved after surgery [34]. Single support phases of the two limbs were also progressively more similar over time. These aspects indicate that the amputee limb avoidance that occurred with the socket-type prosthesis was almost entirely absent with the osseointegrated prosthesis. Therefore, the patient was more confident in loading the amputee limb during gait. The perception of an “own limb” is one of the advantages claimed by patients wearing an osseointegrated prosthesis [35]. Such confidence allows the patient to wear the prosthesis more hours per day and prolong the time spent walking at a physiological speed. The asymmetrical preoperative gait is in line with several previous studies [36,37,38,39]. In particular, Cutti et al. indicated a limb symmetry index of 1.11–1.22 for transfemoral amputee patients, 1.03 for transtibial amputee patients, and of 1.02 for healthy controls [23]. Moreover, a coefficient of variation lower than 2.6% has been associated with a normal gait [24], while previous studies on transfemoral amputee patients with a bone-anchored prosthesis reported an average coefficient of variation of 8.8% [18]. Thus, the symmetry index and coefficient of variation of the patient involved in the present study were closer to normality than transfemoral amputee patients wearing socket-type and closer to transtibial amputee patients and healthy controls, especially at one-year follow-up [23]. The recovery of a symmetrical gait is of crucial importance for a patient’s long-term physical function since it reduces the risk of gait compensations, muscle impairments, and bone loss [31,40]. Symmetrical gait is also fundamental for a patient’s acceptance of the amputee condition since it allows him/her to move without an evident handicap.

Significant joint kinematics improvement was noted at the hip and pelvis levels pre-op to post-surgery. The patient landed with a progressively decreased hip flexion and extended more at the toe-off (Figure 4 and Figure 5). These aspects suggest a lower distance between the center of mass and the center of pressure during the entire stance phase and a larger propulsion with both limbs at the terminal stance. The greater hip mobility shown by the patient could be related to the longer time and load spent on the amputee limb after osseointegration surgery and contributes to improving gait efficacy [31]. Pelvis anteversion also progressively decreased through the follow-ups, indicating a more upright upper body posture. Hip and pelvis kinematics on the sagittal plane were extremely similar between the last two follow-ups (6M and 1Y), suggesting a stabilization of the gait pattern over time. Both pelvis and trunk kinematics have a crucial role in transfemoral amputee patients. The improved hip mobility on the sagittal plane and the reduced pelvis anteversion contributed to reducing the frontal and transverse plane range of motion. Thus, limited limping and better directional control could be inferred. People with transfemoral amputation generally adopt abnormal gait strategies to compensate for deficits in muscle strength and joint mobility. In particular, a reduced range of motion at the sagittal plane and increased frontal and transverse plane rotations at the hip and pelvis joints are reported [3]. Previous studies showed reduced hip flexion and extension in the residual limb and large hip abduction and rotation as compensatory movements during walking at a self-selected speed [30,41].

Compensatory patterns including poor hip muscle control and insufficient ankle dorsiflexion were also reported and are supposed to affect the stabilization of the pelvis during the stance phase, increasing pelvic obliquity [30,42,43]. Such compensatory strategies increase the risks of low back pain and muscle imbalance, which are recurrent concerns for individuals with lower limb amputations. Moreover, degenerative changes at intact joints, in particular hip osteoarthritis of the sound limb, could be worsened by such compensations [34,44,45,46]. The modifications in gait kinematics found in the present study after osteointegration surgery suggest an overall improved movement quality and a reduced risk of the comorbidities usually associated with the transfemoral amputee condition. The patient demonstrated greater motor control despite maintaining a good speed and cadence compared to the socket-type prosthesis: during the last follow-up, the hip abduction increased in response to the greater hip extension. However, the pelvic balance was not compromised, and compensatory strategies did not emerge.

The Q-TFA results pointed out serious deficits in prosthetic use and prosthesis-related problems compared to the reference values in the pre-operative phase [21]. After just 3M, each sub-score sensibly improved and continued until a 1-year follow-up. The assessment of the pre-surgery EQ-5D-5L showed that the patient had the most severe problems in terms of mobility, usual activities, and pain/discomfort. In a prospective case-control study on a large population with socket-type that moved to osseointegrated prosthesis, Van de Meent et al. and Hagberg et al. also reported a Q-TFA score of 39 and 38 with the socket (our study: 33) and a significant improvement at 1-year and 5–10-years follow-up (63–74 points; our study: 92) [3,47]. The results of the present study are also comparable to the study conducted by Ernstssoner et al. [27]. After 3M, mobility and usual activities results decreased until reaching the minimum value, and pain and discomfort also strongly decreased even if not reaching the optimal value.

The longitudinal assessment provided in the present case study demonstrated a clinically relevant improvement in gait functionality after the osseointegration surgery. A more physiological load transmission at the amputee limb could be inferred. The advantage of the osseointegrated prosthesis in amputee patients that have difficulties in wearing a socket-type could be therefore not limited to the avoidance of phantom limb perception and skin sweating but could be visible also in long-term disease reduction. A relatively short adaptation period was noted, with significant differences already visible between the pre-op and the first follow-up. The use of wearables allowed gait analysis in a more familiar environment (hospital hall) and has room for future studies in outdoor or home locations.

The present study has several limitations. First, it represents the analysis of a single case undergoing osseointegrated surgery and rehabilitation within a research setting. Thus, results should be generalized with caution. A transfemoral amputation could be necessary for a multitude of reasons and could lead to very different outcomes and changes in a person’s life. The patient enrolled in the present study was physically active and well-versed in undergoing the treatment. Despite the efficacy of the surgical procedure itself, the physical and psychological readiness of the patient are fundamental for the long-term success of the treatment. Few osseointegration surgeries are performed worldwide compared to the number of amputations. However, the absolute numbers are increasing and there might be room for future larger cohort studies and longer follow-ups. Second, no ground reaction forces were collected. Thus, it was not possible to assess joint dynamics. Such an analysis could be of interest as the input for numerical simulations and finite element modeling of daily life loads at the bone-prosthesis interface. Load sensors inside the prosthesis have also the potential to offer precious information regarding the quality and durability of an implant: a previous study reported up to 100% BW carried by the osseointegrated prosthesis (press-fit design) during walking [48]. Furthermore, the use of insole sensors could have provided valuable information on pressure distribution over the gait. Third, gait speed was not standardized during the follow-ups. The patient was asked to walk at a self-selected speed and the maximum speed possible. However, gait speed did not change over the follow-ups and did not differ from the preoperative condition. This was possible due to the active physical status of the patient. Last, the stump was long enough to allow for a consistent sensor positioning at the thigh between the socket-type and the osseointegrated conditions. However, no full-limb setup for prosthetic limbs exists for wearable sensors. This should be the object of future research to improve the quality and consistency of data analysis from wearable sensors technology.

## 5. Conclusions

Spatiotemporal and gait kinematics improved after osseointegration surgery in a patient with transfemoral amputation compared to a standard socket-type prosthesis. Symmetry indices one year after surgery were close to non-pathological gait and gait compensation sensibly decreased. From a functional point of view, osseointegration surgery could be a valid solution in patients with transfemoral amputation facing issues with traditional socket-type prostheses. The present case study has the potential to inform the design of future larger studies focusing on practical and relevant clinical assessments of walking abilities in amputee patients. The use of wearable sensors in out-of-lab environments and the analysis through Statistical Parametric Mapping can improve the ecological validity and interpretation of patients’ functionalities.

## Figures and Tables

**Figure 1 sensors-23-04037-f001:**
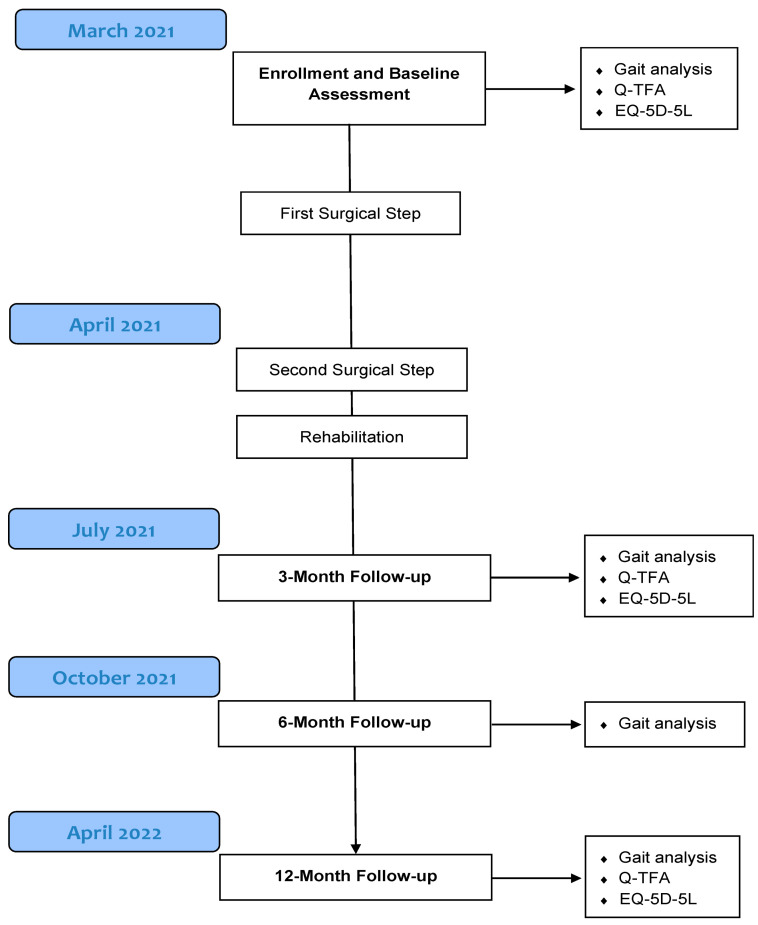
Flow diagram presenting the longitudinal assessment of the transfemoral amputee patient. The surgical procedure is presented in detail in Section 2.2.

**Figure 2 sensors-23-04037-f002:**
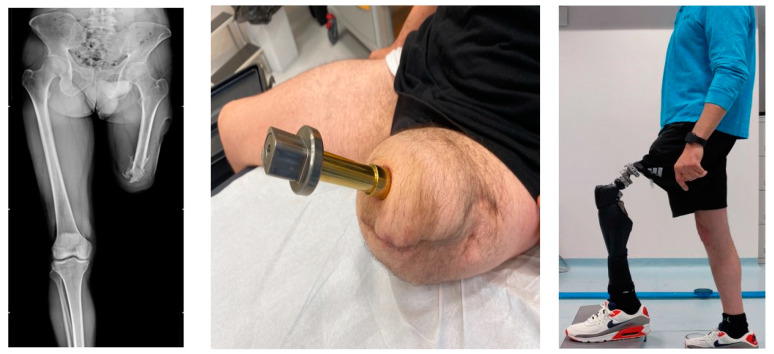
X-ray front view (**left**), bone-prosthesis connection after the second surgical step (**middle**), patient wearing the prosthesis at one-year follow-up (**right**).

**Figure 3 sensors-23-04037-f003:**
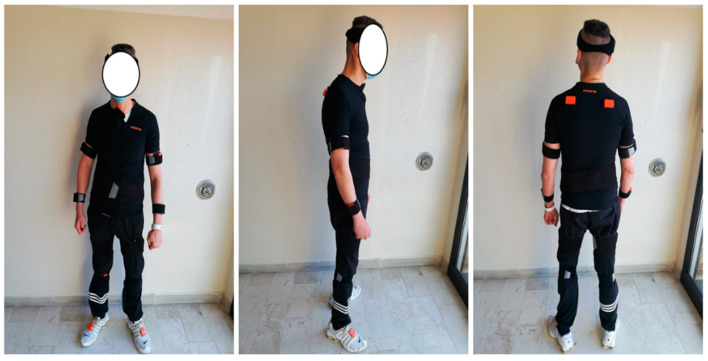
Sensors placement. Full-body placement of inertial sensor units for gait analysis in front view (**left**), side view (**middle**), and back view (**right**). Note: the sensors on the thigh were placed on the residual stump for the amputee leg and symmetrically on the sound limb, according to the manufacturer’s instruction.

**Figure 4 sensors-23-04037-f004:**
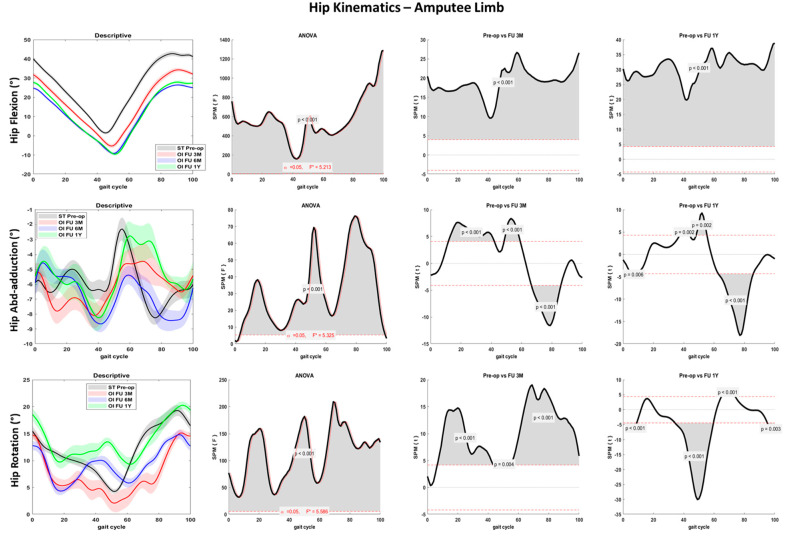
Hip joint kinematics for the amputee limb over the percentage of the gait cycle for the fast gait in sagittal (first row), frontal (second row), and transverse (third row) planes. Descriptive data (first column) are expressed as mean (solid line) and standard deviation (dashed line). The gray line represents pre-operative gait analysis with socket-type prosthesis (ST pre-op), the red line represents gait analysis at 3 months follow-up with osseointegrated prosthesis (OI FU 3M), the blue line represents gait analysis at 6 months follow-up with osseointegrated prosthesis (OI FU 6M), the green line represents gait analysis at 1-year follow-up with osseointegrated prosthesis (OI FU 1Y). The results of the spm1D repeated-measure ANOVA are presented in the second column. Gray areas represent statistically significant differences (*p* < 0.05) among the groups. The differences between pre-operative phase and either 3 months follow-up (third column) or 1-year follow-up (fourth column) were assessed through t-test with Bonferroni correction.

**Figure 5 sensors-23-04037-f005:**
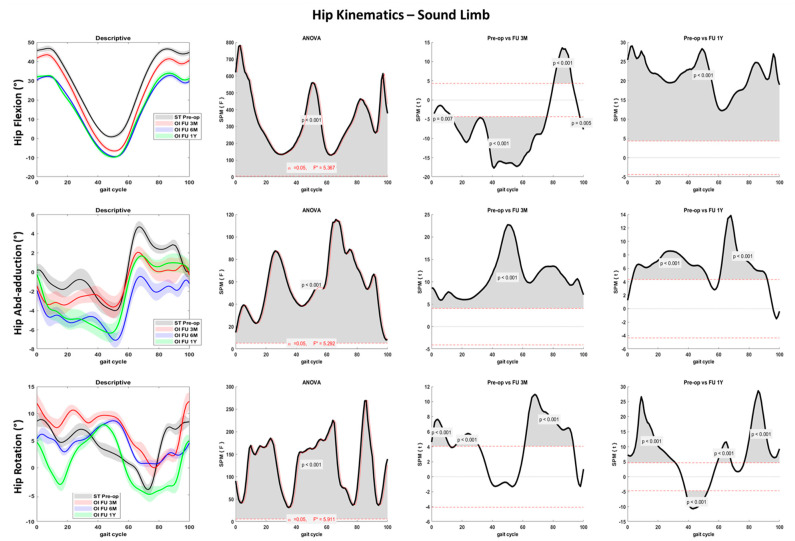
Hip joint kinematics for the sound limb over the percentage of the gait cycle for the fast gait in sagittal (first row), frontal (second row), and transverse (third row) planes. Descriptive data (first column) are expressed as mean (solid line) and standard deviation (dashed line). The gray line represents pre-operative gait analysis with socket-type prosthesis (ST pre-op), the red line represents gait analysis at 3 months follow-up with osseointegrated prosthesis (OI FU 3M), the blue line represents gait analysis at 6 months follow-up with osseointegrated prosthesis (OI FU 6M), the green line represents gait analysis at 1-year follow-up with osseointegrated prosthesis (OI FU 1Y). The results of the spm1D repeated-measure ANOVA are presented in the second column. Gray areas represent statistically significant differences (*p* < 0.05) among the groups. The differences between pre-operative phase and either 3 months follow-up (third column) or 1-year follow-up (fourth column) were assessed through t-test with Bonferroni correction.

**Figure 6 sensors-23-04037-f006:**
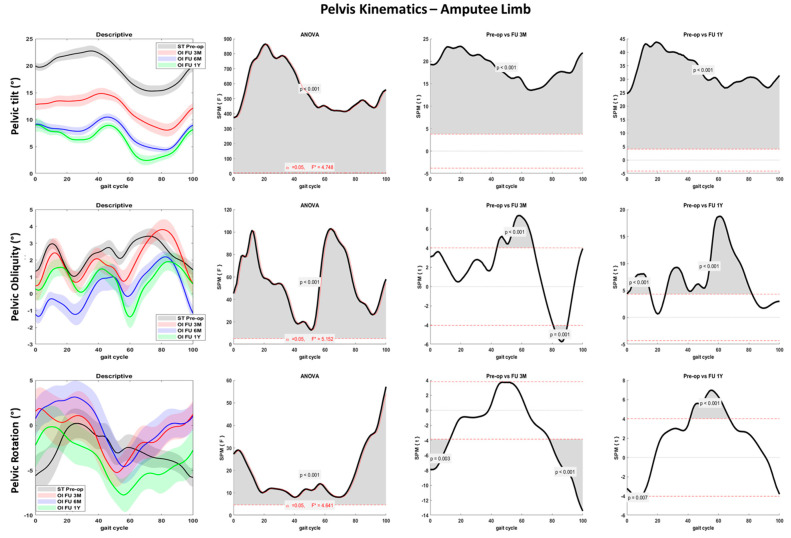
Pelvis joint kinematics over the percentage of gait cycle performed with the amputee limb for the fast gait in sagittal (first row), frontal (second row), and transverse (third row) planes. Descriptive data (first column) are expressed as mean (solid line) and standard deviation (dashed line). The gray line represents pre-operative gait analysis with socket-type prosthesis (ST pre-op), the red line represents gait analysis at 3 months follow-up with osseointegrated prosthesis (OI FU 3M), the blue line represents gait analysis at 6 months follow-up with osseointegrated prosthesis (OI FU 6M), the green line represents gait analysis at 1-year follow-up with osseointegrated prosthesis (OI FU 1Y). The results of the spm1D repeated-measure ANOVA are presented in the second column. Gray areas represent statistically significant differences (*p* < 0.05) among the groups. The differences between pre-operative phase and either 3 months follow-up (third column) or 1-year follow-up (fourth column) were assessed through t-test with Bonferroni correction.

**Figure 7 sensors-23-04037-f007:**
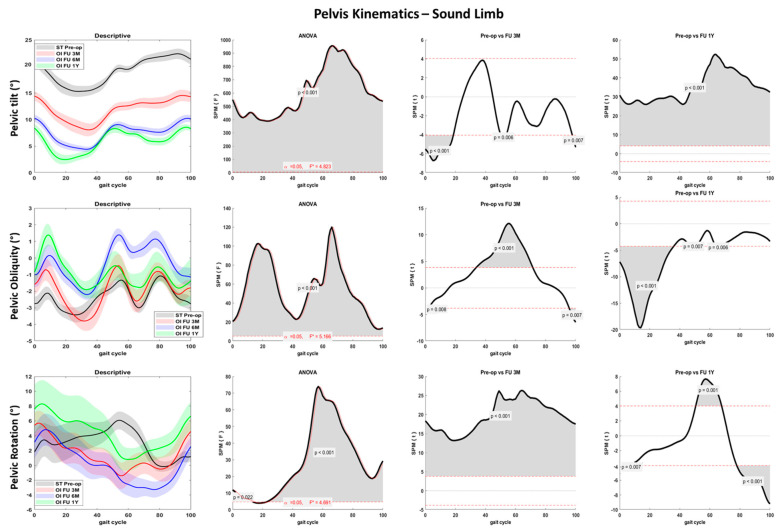
Pelvis joint kinematics over the percentage of gait cycle performed with the sound limb for the fast gait in sagittal (first row), frontal (second row), and transverse (third row) planes. Descriptive data (first column) are expressed as mean (solid line) and standard deviation (dashed line). The gray line represents pre-operative gait analysis with socket-type prosthesis (ST pre-op), the red line represents gait analysis at 3 months follow-up with osseointegrated prosthesis (OI FU 3M), the blue line represents gait analysis at 6 months follow-up with osseointegrated prosthesis (OI FU 6M), the green line represents gait analysis at 1-year follow-up with osseointegrated prosthesis (OI FU 1Y). The results of the spm1D repeated-measure ANOVA are presented in the second column. Gray areas represent statistically significant differences (*p* < 0.05) among the groups. The differences between pre-operative phase and either 3 months follow-up (third column) or 1-year follow-up (fourth column) were assessed through t-test with Bonferroni correction.

**Figure 8 sensors-23-04037-f008:**
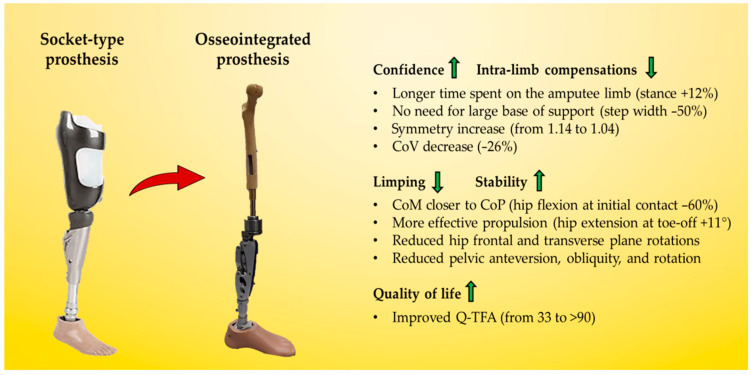
Summary of the functional and quality of life improvements in the patient from socket-type prosthesis to one-year after osseointegrated prosthesis surgery.

**Table 1 sensors-23-04037-t001:** Anthropometric parameters of the transfemoral amputee patient case.

Anthropometric Parameters	
Mass (kg)	67.0
Height (m)	1.77
BMI	21.4
Time from amputation at surgery (y)	18
Knee prosthesis	Genium X3, (Ottobock SE & Co. KGaA, Duderstadt, Germany)
Ankle prosthesis	Passive Mechanism
Sound limb length (cm)	53.3
Amputee limb length (cm)	26.5
Amputee limb—percentage of sound thigh (%)	49.7

Note: BMI: Body Mass Index.

**Table 2 sensors-23-04037-t002:** Spatial gait parameters for the pre-operative gait analysis (socket-type prosthesis, ST) and for the follow-up gait analyses (osseointegrated prosthesis, OI). Data are presented for the fast gait as mean ± standard deviation. Note: “Difference” refers to a Sound limb—Amputee limb variable.

Spatial Parameters		Pre-Op	Follow-Up 3M	Follow-Up 6M	Follow-Up 1Y
Speed (m/s)		1.63 ± 0.03	1.66 ± 0.04	1.55 ± 0.03	1.68 ± 0.02
Cadence (steps/min)		131.67	127.67	125.97	130.43
Step Length (cm)	Amputee Limb	75.08 ± 2.72	71.42 ± 2.00	74.28 ± 1.59	69.23 ± 1.82
	Sound Limb	82.65 ± 1.83	88.20 ± 1.76	74.92 ± 1.42	86.11 ± 2.47
	Difference	−7.56	−16.78	−0.64	−16.88
Step Width (cm)	Amputee Limb	13.74 ± 1.81	8.28 ± 1.70	4.71 ± 1.35	6.66 ± 2.20
	Sound Limb	14.19 ± 1.10	8.84 ± 1.98	4.99 ± 1.27	7.31 ± 4.43
	Difference	−0.45	−0.56	−0.28	−0.66

**Table 3 sensors-23-04037-t003:** Temporal gait parameters for the pre-operative gait analysis (socket-type prosthesis, ST) and the follow-up gait analyses (osseointegrated prosthesis, OI). Data are presented for the fast gait as mean ± standard deviation as a percentage of the gait cycle. Note: “Difference” means Sound limb—Amputee limb variable; symmetry index = 1.00 means perfect symmetry between stance phases of the two limbs.

Temporal Parameters (% of Gait Cycle)		Pre-Op	Follow-Up 3M	Follow-Up 6M	Follow-Up 1Y
Stance	Amputee Limb	51.00 ± 0.98	54.82 ± 1.30	57.13 ± 1.34	57.23 ± 0.77
	Sound Limb	58.15 ± 0.81	59.62 ± 0.97	60.53 ± 0.78	59.35 ± 1.25
	Difference	−7.15	−4.8	−3.4	−2.12
Swing	Amputee Limb	49.00 ± 0.98	45.18 ± 1.30	42.87 ± 1.34	42.77 ± 0.77
	Sound Limb	41.71 ± 1.14	40.27 ± 1.35	39.59 ± 1.29	40.92 ± 1.21
	Difference	7.3	4.91	3.28	1.85
Symmetry Index		1.14	1.09	1.06	1.04
Coefficient of Variation	Amputee Limb	1.97	2.48	1.78	1.13
	Sound Limb	1.50	1.79	1.29	1.19
	Difference	−0.47	−0.70	−0.48	0.06
Single Support	Amputee Limb	41.90 ± 1.27	40.27 ± 1.06	39.43 ± 1.06	40.82 ± 1.12
	Sound Limb	48.79 ± 1.38	45.17 ± 0.97	43.04 ± 1.25	42.88 ± 1.17
	Difference	−6.89	−4.9	−3.61	−2.06
Double Support	Amputee Limb	3.03 ± 1.13	7.60 ± 1.18	9.08 ± 0.99	9.74 ± 0.50
	Sound Limb	6.04 ± 1.14	6.96 ± 0.90	8.66 ± 0.97	6.68 ± 1.20
	Difference	−3.01	0.64	0.43	3.06
	Total	9.07 ± 1.82	14.56 ± 1.61	17.74 ± 1.65	16.42 ± 1.16

**Table 4 sensors-23-04037-t004:** Descriptive peak hip joint kinematics for the amputee and sound limb in the pre-operative gait analysis (socket-type prosthesis, ST) and the follow-up gait analyses (osseointegrated prosthesis, OI). Data are presented for the fast gait as mean ± standard deviation separately for stance and swing phases.

Hip Kinematics		Pre-Op	Follow-Up 3M	Follow-Up 6M	Follow-Up 1Y
Hip Flexion (°)					
Amputee Limb	Stance Max	40 ± 0.8	31.8 ± 1.2	24.8 ± 0.6	27.7 ± 1
	Stance Min	1.4 ± 0.8	−5.5 ± 1	−9.4 ± 0.8	−9.6 ± 0.8
	Swing Max	42.8 ± 1.2	34.4 ± 1	26.5 ± 0.4	28 ± 0.8
	Swing Min	4.9 ± 1.2	−1.7 ± 1.5	−4.8 ± 1.8	−6.2 ± 1
Sound Limb	Stance Max	47 ± 1.3	43.7 ± 1.2	32.3 ± 0.7	33.1 ± 0.5
	Stance Min	0.8 ± 0.8	−6.6 ± 0.8	−9.5 ± 0.6	−9.8 ± 0.7
	Swing Max	46.7 ± 1	41.6 ± 1.4	32.8 ± 1.1	34.2 ± 1.2
	Swing Min	1 ± 0.9	−5.8 ± 0.8	−7.7 ± 0.7	−7.1 ± 1.2
Hip Abd-adduction (°)					
Amputee Limb	Stance Max	−4.2 ± 0.7	−4.7 ± 0.8	−4.4 ± 0.9	−3.5 ± 0.9
	Stance Min	−6.8 ± 0.6	−8.4 ± 0.7	−8.7 ± 0.6	−8.3 ± 0.9
	Swing Max	−2.3 ± 0.8	−4.1 ± 1	−5.1 ± 0.6	−2.6 ± 1
	Swing Min	−8.3 ± 0.5	−6.7 ± 0.4	−8.7 ± 0.6	−6.5 ± 0.8
Sound Limb	Stance Max	0.4 ± 0.6	−1.2 ± 0.9	−1.9 ± 1	−0.3 ± 1
	Stance Min	−4 ± 0.8	−4.3 ± 0.7	−7.1 ± 0.8	−6.4 ± 0.9
	Swing Max	4.7 ± 0.6	2.1 ± 0.6	−0.2 ± 0.9	2 ± 0.4
	Swing Min	−4.1 ± 0.8	−3.3 ± 0.9	−6.1 ± 0.8	−4 ± 1
Hip Rotation (°)					
Amputee Limb	Stance Max	15.5 ± 1	14.8 ± 0.8	13 ± 0.8	18.5 ± 1.1
	Stance Min	4.5 ± 0.5	2 ± 1.8	4.2 ± 0.7	9.6 ± 1.2
	Swing Max	19.4 ± 0.9	15.4 ± 0.7	14.9 ± 0.8	20.3 ± 0.7
	Swing Min	4.2 ± 0.4	2.3 ± 1.9	5.8 ± 0.8	9.1 ± 1.3
Sound Limb	Stance Max	9.2 ± 1.2	12.2 ± 1.5	8.7 ± 0.5	8.1 ± 0.3
	Stance Min	2.7 ± 1.2	7 ± 1.1	2.8 ± 0.7	−3.3 ± 1.3
	Swing Max	9.1 ± 0.8	12.4 ± 1.7	7.4 ± 0.7	5.3 ± 1.1
	Swing Min	−4.3 ± 0.9	−0.3 ± 1.5	0.5 ± 0.5	−5.2 ± 1.2

**Table 5 sensors-23-04037-t005:** Descriptive peak pelvis kinematics for the amputee and sound limb in the pre-operative gait analysis (socket-type prosthesis, ST) and the follow-up gait analyses (osseointegrated prosthesis, OI). Data are presented for the fast gait as mean ± standard deviation separately for stance and swing phases.

Pelvis Kinematics		Pre-Op	Follow-Up 3M	Follow-Up 6M	Follow-Up 1Y
Pelvis Tilt (°)					
Amputee Limb	Stance Max	0.3 ± 1.7	2.5 ± 2.6	3.6 ± 1.7	0.2 ± 2.1
	Stance Min	−5.6 ± 2.2	−5.4 ± 1.7	−4.6 ± 1.9	−7.8 ± 1.8
	Swing Max	−2.2 ± 1.3	1.2 ± 1.5	1.1 ± 1.6	−2.8 ± 2.5
	Swing Min	−5.8 ± 1	−5 ± 1.6	−4.7 ± 1.9	−7.8 ± 1.9
Sound Limb	Stance Max	5.7 ± 1.3	5.8 ± 1.8	4.9 ± 2.2	8.4 ± 3.2
	Stance Min	1.7 ± 2.5	−1.3 ± 1.9	−2.2 ± 1.6	1 ± 1.7
	Swing Max	6.2 ± 1.2	4.6 ± 1.7	2.6 ± 1.3	6.6 ± 1.9
	Swing Min	−0.3 ± 0.7	−1.9 ± 1.7	−3.7 ± 1	0.4 ± 1.6
Pelvis Obliquity (°)					
Amputee Limb	Stance Max	3.1 ± 0.3	2.6 ± 0.8	1.1 ± 0.7	1.7 ± 0.5
	Stance Min	0.9 ± 0.4	0.2 ± 0.6	−1.6 ± 0.5	−0.7 ± 0.6
	Swing Max	3.5 ± 0.4	3.8 ± 0.6	2.2 ± 0.3	1.9 ± 0.5
	Swing Min	1.4 ± 0.3	0.2 ± 0.5	−1.2 ± 0.4	−1.4 ± 0.7
Sound Limb	Stance Max	−1.6 ± 0.3	−0.1 ± 0.5	1.4 ± 0.4	1.4 ± 0.7
	Stance Min	−3.5 ± 0.4	−3.8 ± 0.6	−2.2 ± 0.3	−1.9 ± 0.5
	Swing Max	−1 ± 0.4	−0.3 ± 0.5	1.4 ± 0.3	−0.4 ± 0.8
	Swing Min	−3.1 ± 0.3	−2.7 ± 0.7	−1.2 ± 0.5	−2 ± 1
Pelvis Rotation (°)					
Amputee Limb	Stance Max	22.8 ± 1	14.9 ± 1.1	10.5 ± 0.6	9.4 ± 1
	Stance Min	19.6 ± 0.7	12.6 ± 0.9	7.8 ± 0.7	6.2 ± 0.6
	Swing Max	20.2 ± 0.9	13.4 ± 0.8	9.1 ± 0.6	8.2 ± 0.9
	Swing Min	15.3 ± 1	8 ± 1.2	4.4 ± 0.5	2.4 ± 0.9
Sound Limb	Stance Max	21.3 ± 1	14.4 ± 0.9	10.3 ± 0.6	8.6 ± 0.8
	Stance Min	15.3 ± 1	8 ± 1.2	4.4 ± 0.5	2.4 ± 0.9
	Swing Max	22.4 ± 1	14.6 ± 1.1	10.3 ± 0.7	8.6 ± 0.5
	Swing Min	18.9 ± 0.6	12.3 ± 0.8	7.6 ± 0.7	5.7 ± 0.7

**Table 6 sensors-23-04037-t006:** Q-TFA values for the pre-operative and follow-up related quality of life evaluation for a person with transfemoral amputation. The data range is from 0 to 100.

Q-TFA	Pre-Op	Follow-Up 3M	Follow-Up 1Y
Prosthetic Use	51.6	90.3	100
Prosthetic Mobility	75.5	93.3	92.2
Problem	47.5	13.3	5.4
Global	33.3	91.6	100

**Table 7 sensors-23-04037-t007:** EQ-5D-5L values for the pre-operative and follow-up related quality of life evaluation. The five domains’ data range is from 1 to 5. General Health is represented by a visual analog scale range from 0 to 100.

EQ-5D-5L	Pre-Op	Follow-Up 3M	Follow-Up 1Y
Mobility	3	1	1
Self-care	2	1	1
Usual Activities	3	1	1
Pain/Discomfort	4	2	2
Anxiety/Depression	1	1	1
General Health	40	75	80

## Data Availability

Not applicable.

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
