# Peer review of "Longitudinal Gait Analysis of a Transfemoral Amputee Patient: Single-Case Report from Socket-Type to Osseointegrated Prosthesis"

_sensors, 2023, doi:10.3390/s23084037_

Round 1

Reviewer 1 Report

1.  Make a  declaration in the manuscript that no humans or animals were harmed during the dataset collection.

2. The statistical analysis can be more extensive. Authors can present some more parametric analysis and can present using a tabular. 

3. The work presented is quite interesting, and is complete in every aspect, however for better reader's understanding author's should make some changes. It is my suggestion to draw a flow diagram for your proposed work.

4. Use some pictorial representation for better understanding the outcome of the results. Like how much percentage the amputee recovered. Add Some graphical representation.

3. Read the following paper for alternative method of gait analysis.

Bijalwan, V., Semwal, V. B., & Mandal, T. K. (2021). Fusion of multi-sensor-based biomechanical gait analysis using vision and wearable sensor. IEEE Sensors Journal21(13), 14213-14220.

Reviewer 2 Report

Very interesting work that provides a case study. It is understood that each situation is very individual and we often say that a sample is needed, that the same factors will not necessarily work, and so on. And both sides are right. In this case, the situation is exceptional and deserves a case study. The aim of the paper was to provide a longitudinal functional assessment of a patient with transfemoral amputation from the preoperative status with a socket-type prosthesis to one year after the osseointegration surgery. Detailed and clear spatiotemporal parameters, joint kinematics, and self-evaluation results of this study are presented.

Maybe a few more trivial suggestions. It would be interesting to know the total sample of data that was subjected to statistical significance assessment. Also, the presented results of different groups can be significantly correlated in the light and thus indicate valuable guidelines for function evaluation. In the conclusions, I would present several results that support the statements.

The results are presented very clearly and visually using modern tools like SPM. Seeing how the authors themselves evaluate their research and how they present the shortcomings, I do not want to comment further. The work is very interesting, detailed and clear.

Reviewer 3 Report

Relevance of this work

The authors are addressing a genuine and important clinical question: What are the differences in walking abilities and self-reported functional outcomes moving from socket-suspended to bone-anchored prosthesis? The authors provide examples of these outcomes through a single case. However, they presented some unique datasets, particularly the kinematic data measured with inertial sensors (IMU).

Clearly, there is a need for this type of publication that could play an important role in informing the design of future studies focusing a practical and relevant clinical assessment of walking abilities with transfemoral bone-anchored prostheses.   

Limitations of this work

Below, you will find a list of approximately 30 comments including minor suggestions and major changes. In my opinion, the manuscript can gain in thoroughness and clarity if it follows the STROBE guideline for case report (see: https://www.equator-network.org/reporting-guidelines/strobe/).  

Altogether, this work requires critical adjustments in the following sections:

  • Introduction including:
    • Add small section describing the need for this study based on the limitations of previous studies focusing on walking abilities and self-reported functional outcomes,
    • Add small section highlight the originality of this work and rationales for the study design including new approach using IMU and potential way at the light of exciting evaluation framework.
  •  Methods including:
    • Add typical confounders such mass, height, body mass index, length of the residuum in centimeters and percentage of sound thigh, type of knee and ankle/foot components, static and dynamic alignment information as well as number of gait cycles analysed. 
  • Results including:
    • Add all the datasets including the ones for the self-selected speed,
  • Discussion including:
    • Temper several strong statements partially backed up by the data presented and possibly biased by the focus of fast gait data,
    • Put the proposed outcomes in contrast with previous baseline studies focusing on walking abilities and self-reported functional outcomes,
    • Highlight how the use of IMU and SPM1D can improve the design of evaluation frame and assessments of walking abilities.
  • Conclusion including:
    • Highlight how this particular study informs the design of future studies focusing on practicaland relevant clinical assessments of walking abilities.
  • References including:
    • Consider and cite recent key references focusing particularly on participants with transfemoral bone-anchored prostheses.

Outcomes

Despite these concerns, I do believe that the core work of the study can be original and worthwhile publishing provided that the authors address the points above. I am confident that a substantial publication that will be well-cited could come out of this work.

Selected specific comments

Title

1.    L 1: Consider adding the study design (e.g., single case report) in the title.

Introduction

2.    L 37: Clarify how Ref 2 focusing on the “water resource management” is relevant.

3.    L 65: Consider and cite the references related to hip and knees kinetic (1, 2) and spatio-temporal gait parameters (3, 4, 5, 6) for individuals fitted with transfemoral bone-anchored prostheses. It will be relevant to discuss some of the references and highlight their limitations, thereby, the need for this study.

4.    L 66: Temper this statement discussing and citing key reference (4).  

5.    L 66: Add a few sentences about the rationales for conducting this particularly study at the light of current evaluation frame to assess the efficacy and safety of bone-anchored prostheses. I will suggest highlighting the need to collect kinematic data with easy-to-use and clinically relevant IMU sensors (7).   

Methods

6.    L 74: Replace “transfemoral amputee patient” by “a participant with transfemoral amputation”.

7.    L 110: Add typical confounders to facilitate subsequent meta-analyses including mass, heigh, body mass index, length of the residuum in centimeters and percentage of sound thigh, type of knee and ankle/foot components as well as static and dynamic alignment information.

8.    L 116: Provide the brand and model of the implant.

9.    L 138: Start the sentence with “Three”.  

10.  L 180: Clarify if the gait events such as heel contacts and toe-offs were automatically identified using the Gait Analysis Report in the Xsens Motion Cloud or manually through the customized script in Matlab.

11.  L 189: Clarify the process used to reduce noise in kinematic data (e.g., smoothing, filtering).

Results

12.  L 212: Add all the results for the self-selected speed in Tables and Figure to facilitate the subsequent meta-analyses. I can understand that the authors want to focus the manuscript only on the variable showing differences. By doing so, they also induce a bias. I believe finding that minimal differences in terms of speed and kinematics emerged between self-selected speed gait and fast gait is a critical outcome. These datasets gathered at self-selected speed are important to establish “baseline”.

13.  L 212: Provide the total number of gait cycles analyzed in all conditions.

Discussion

14.  L 320: Temper this statement because it is only true when considering fast gait, as mentioned in the first sentences of the Results.  

15.  L 340: Consider and cite references values extracted from the literature focusing on spatio-temporal gait parameters for transfemoral bone-anchored prostheses (3, 4, 5)

16.  L 351: Consider and cite references focusing on loading on transfemoral bone-anchored prosthesis (8).

17.  L 384: Consider and cite the references related to hip and knees kinetic (1, 2).

18.  L 425: Mention other longitudinal studies that have been on-going for over a decade (9)

19.  L 429: Add comment about the opportunity to insert load sensors inside the prosthesis that can provide better information about load distribution and feed input data for finite element models (8, 10).

20.  L 437: Highlight how the use of IMU and SPM1D can improve the design of evaluation frame, particularly the assessment of ambulation and walking abilities described in Figure 1 of (7).   

Conclusions

21.  L 438: Highlight how this particular study informs the design of future studies focusing on practical and relevant clinical assessments of walking abilities (e.g., use of IMU, range of differences to determine statistical power and sample size).

Illustrations

22.  Tables 1-2, Figures 3-6: Indicate that the results are for the fast gait conditions.

23.  Tables 1-2, Figures 3-6 Consider adding all datasets including self-selected speed.

24.   Figures 2-6: Increase the size of the font as the captions are generally difficult to read as formatted.

References

25.  L 459: Consider including key references focusing on participants with transfemoral bone-anchored prosthesis.

26.  Ref 2: Consider deleting this reference focusing on “water resource management”.

27.  L 560: Consider discussing and citing the following references:

1.         Dumas R, Branemark R, Frossard L. Gait analysis of transfemoral amputees: errors in inverse dynamics are substantial and depend on prosthetic design. IEEE transactions on neural systems and rehabilitation engineering : a publication of the IEEE Engineering in Medicine and Biology Society. 2017;25(6):679-85.

2.         Harandi VJ, Ackland DC, Haddara R, Cofre Lizama LE, Graf M, Galea MP, et al. Individual muscle contributions to hip joint-contact forces during walking in unilateral transfemoral amputees with osseointegrated prostheses. Comput Methods Biomech Biomed Engin. 2020:1-11.

3.         Frossard L, Hagberg K, Häggström E, Gow DL, Brånemark R, Pearcy M. Functional Outcome of Transfemoral Amputees Fitted With an Osseointegrated Fixation: Temporal Gait Characteristics. JPO Journal of Prosthetics and Orthotics. 2010;22(1):11-20.

4.         Van de Meent H, Hopman MT, Frolke JP. Walking ability and quality of life in subjects with transfemoral amputation: a comparison of osseointegration with socket prostheses. Arch Phys Med Rehabil. 2013;94(11):2174-8.

5.         Gailey R, Lucarevic J, Clemens S, Agrawal V, Bennett C, Muderis M, et al. A Comparison of Prosthetic Mobility in Amputees with Osseointegration versus Traditional Amputation and Socket.  43rd Academy Annual Meeting & Scientific Symposium of the American Academy of Orthotists & Prosthetists2017. p. FPTH14.

6.         Sinclair S, Beck JP, Webster J, Agarwal J, Gillespie B, Stevens P, et al. The First FDA Approved Early Feasibility Study of a Novel Percutaneous Bone Anchored Prosthesis for Transfemoral Amputees: A Prospective One-year Follow-up Cohort Study. Archives of Physical Medicine and Rehabilitation. 2022.

7.         Berg D, Frossard L. Health service delivery and economic evaluation of limb lower bone-anchored prostheses: A summary of the Queensland artificial limb service’s experience. Canadian Prosthetics & Orthotics Journal. 2021;4(2): 1-22.

8.         Frossard L, Laux S, Geada M, Heym PP, Lechler K. Load applied on osseointegrated implant by transfemoral bone-anchored prostheses fitted with state-of-the-art prosthetic components. Clin Biomech (Bristol, Avon). 2021;89:105457.

9.         Hagberg K, Ghasemi Jahani SA, Omar O, Thomsen P. Osseointegrated prostheses for the rehabilitation of patients with transfemoral amputations: A prospective ten-year cohort study of patient-reported outcomes and complications. Journal of Orthopaedic Translation. 2023;38:56-64.

10.       Prochor P, Frossard L, Sajewicz E. Effect of the material’s stiffness on stress-shielding in osseointegrated implants for bone-anchored prostheses: a numerical analysis and initial benchmark data. Acta of Bioengineering and Biomechanics. 2020;2020(2):69-81.

Round 2

Reviewer 3 Report

Extent of my review

My review of this resubmission focused essentially on (A) considering the ways the authors have addressed the initial concerns raised during the first review and (B) reading the new manuscript as a new standalone piece.

General Comments

My overall impression is that the authors have carefully read the comments and genuinely made significant efforts to address all comments thoroughly both in the manuscript and the supplementary material. The authors addressed satisfactorily my 27 initial comments.

Outcomes

I believe that the core of this study is original and worthwhile publishing. In my view, the manuscript is ready for publication.